# Assessment of Lab4P Probiotic Effects on Cognition in 3xTg-AD Alzheimer’s Disease Model Mice and the SH-SY5Y Neuronal Cell Line

**DOI:** 10.3390/ijms24054683

**Published:** 2023-02-28

**Authors:** Thomas S. Webberley, Ryan J. Bevan, Joshua Kerry-Smith, Jordanna Dally, Daryn R. Michael, Sophie Thomas, Meg Rees, James E. Morgan, Julian R. Marchesi, Mark A. Good, Sue F. Plummer, Duolao Wang, Timothy R. Hughes

**Affiliations:** 1Systems Immunity Research Institute, School of Medicine, Cardiff University, Cardiff CF14 4XW, UK; 2Cultech Limited, Unit 2 Christchurch Road, Baglan Industrial Park, Port Talbot SA12 7BZ, UK; 3UK Dementia Research Institute, Cardiff University, Maindy Road, Cardiff CF24 4HQ, UK; 4School of Optometry and Vision Sciences, Cardiff University, Maindy Road, Cardiff CF10 4HQ, UK; 5Division of Digestive Diseases, Department of Metabolism, Digestion and Reproduction, Faculty of Medicine, Imperial College London, London SW7 2AZ, UK; 6School of Psychology, Cardiff University, Cardiff CF10 3AT, UK; 7Department of Clinical Sciences, Liverpool School of Tropical Medicine, Liverpool L3 5QA, UK

**Keywords:** Alzheimer’s disease, cognition, neurodegeneration, probiotics

## Abstract

Aging and metabolic syndrome are associated with neurodegenerative pathologies including Alzheimer’s disease (AD) and there is growing interest in the prophylactic potential of probiotic bacteria in this area. In this study, we assessed the neuroprotective potential of the Lab4P probiotic consortium in both age and metabolically challenged 3xTg-AD mice and in human SH-SY5Y cell culture models of neurodegeneration. In mice, supplementation prevented disease-associated deteriorations in novel object recognition, hippocampal neurone spine density (particularly thin spines) and mRNA expression in hippocampal tissue implying an anti-inflammatory impact of the probiotic, more notably in the metabolically challenged setting. In differentiated human SH-SY5Y neurones challenged with β-Amyloid, probiotic metabolites elicited a neuroprotective capability. Taken together, the results highlight Lab4P as a potential neuroprotective agent and provide compelling support for additional studies in animal models of other neurodegenerative conditions and human studies.

## 1. Introduction

The gut microbiota is a complex stable microbial meta-community heavily influenced by host genetics, diet and the environment and this community of microorganisms has a major impact upon host physiology [1]. A bidirectional communication pathway exists between the gut microbiota and the central nervous system (CNS), termed the ‘gut–microbiota–brain axis’, and there is increasing evidence linking aging [2] and metabolic status [3] with changes/disruptions in this pathway. There is also growing support for a strong association between the composition of the gut microbiota and neurodegenerative conditions such as Parkinson’s disease (PD) and Alzheimer’s disease (AD) [4,5].

AD is the most common form of dementia with estimates of 850,000 sufferers in the UK [6] and over 50 million worldwide [7]. AD is a progressive brain disease characterised by the deposition of extracellular amyloid beta (Aβ) plaques and intracellular neurofibrillary (tau) tangles that result in substantial and progressive neuronal loss with age [7]. An early and critical feature of the neurodegenerative process in AD is the loss of the neuronal synaptic connections that results in disruption of brain function and connectivity [8]. Such changes dramatically impact upon a sufferer’s memory [9] and executive function—an umbrella term for verbal reasoning, problem-solving, planning, ability to maintain sustained attention, resistance to interference, multitasking, cognitive flexibility, and coping with novelty [10]. There is currently no cure for established AD, only strategies to target early stage disease [10] or to alleviate symptom severity [11]. The major risk factors for AD are age and genetics and it is now understood that excessive body weight during early/mid-life is linked with cognitive impairment and a doubled risk of developing AD during later life [12]. It has been shown that diet-induced obesity in genetically modified AD-prone mice leads to accelerated disease progression [13,14,15]. The increasing proportion of people over 65 in the population and the growing prevalence of obesity in society [16] highlight the need for strategies to ameliorate the onset and severity of dementia.

One area of research that has recently seen some success in ameliorating disease pathology is dietary supplementation with probiotic bacteria (defined as ‘live microorganisms that when administered in adequate amounts confer a health benefit to the host’ [17]). Probiotics are well recognised for their ability to impart metabolic and immunological benefits to the host [18] and there is growing interest in the potential role/use of probiotic supplementation to combat neurocognitive decline or delay/prevent the onset of AD [19]. Pathological and/or cognitive improvements have been observed in both probiotic treated AD-prone mice [19,20,21,22] and healthy/diseased human cohorts [23,24]. The Lab4P probiotic consortium comprises a combination of lactobacilli and bifidobacteria that has been shown to contribute to neuroprotective effects on human neurons in vitro [25] and improve self-assessed mood rating in a free-living adult population [26]. Lab4P can also impact upon host metabolism evidenced by the inhibition of diet-induced weight gain in C57BL/6J mice wild-type mice [27] and significant weight loss in free-living overweight and obese overtly healthy adults receiving daily supplementation [26,28,29,30]. In addition, clear anti-inflammatory effects have been observed in LDR^-/-^ mice receiving the probiotic [31]. These data imply a potential benefit during neurological disorders although efficacy of Lab4P on AD-related brain pathology has, so far, not been evaluated.

In the current study, we test the hypothesis that Lab4P administered to 3xTg-AD mice (with or without a metabolic challenge) will have neuroprotective effects resulting in improved cognition (novel object memory) and reduce hippocampal neuronal spine density loss. This hypothesis was tested further in human SH-SY5Y neuronal cells challenged with β-Amyloid peptides. Our findings provide novel evidence that dietary supplementation with Lab4P has beneficial effects in mouse and cell models of disease.

## 2. Results

### 2.1. Lab4P Maintains Cognitive Performance and Preserves Hippocampal Spine Densities in 3xTg-AD Mice (Study A)

Male mice (*n* = 54) were fed from birth with standard chow diet (Teklad diet by Envigo, Appendix A) for 24 weeks. The baseline group (BLA; *n* = 18) were then sacrificed while the probiotic group (*n* = 18) received standard chow supplemented with Lab4P (Chow-P) for 24 weeks and the control group (*n* = 18) received standard chow (Chow-C) for 24 weeks (Figure 1A). Calculation of movements of the chow-fed 3xTg-AD mice during Novel Object Recognition (NOR) testing (Figure 1Bi) revealed differences in discrimination ratio (DR) between the Chow-C and Chow-P groups (Figure 1Bii) after 12 weeks (35.2%, *p* = 0.0198) and 24 weeks (40.8%, *p* = 0.317) in the intervention favouring the probiotic. The DR at baseline was 0.71 ± 0.02 and for the Chow-P group at 12 weeks, DR was 0.73 ± 0.04 and at 24 weeks it was 0.69 ± 0.05. The Chow-C group dropped significantly from BLA to 0.54 ± 0.05 (*p* = 0.009) at 12 weeks and 0.49 ± 0.08 (*p* = 0.0321) at 24 weeks.

In the open field test (OFT, Figure 1Biii), no significant between-group differences were observed in total distance travelled, time in the arena centre nor the frequency of rearing, digging or grooming over the course of the study (although Chow-P results were universally higher than Chow-C after 24 weeks supplementation with a trend toward significance for digging: 84.8%, *p* = 0.0847). The behavioural characteristics of the Chow-P group remained consistent with BLA for the entirety of the intervention period, other than digging which increased significantly by 24 weeks (63.7%, *p* = 0.0457). In the Chow-C group, there were significant decreases in time in arena centre at 12 and 24 weeks (−36.9%, *p* = 0.0133 and −49.3%, *p* = 0.0195), respectively, and rearing at 24 weeks (−60.8%, *p* = 0.0145). The temporal changes in DR and OFT results for chow-fed 3xTg-AD mice over the 12-month study period are shown in Appendix A.

To determine whether Lab4P offered a beneficial impact on synaptic plasticity, we measured CA1 dendritic spines from chow-fed 3xTg-AD mice (Figure 2i). Between-group differences in spine loss were seen at 12 weeks (27.1%, *p* < 0.0001, Figure 2ii) and 24 weeks (23.0%, *p* = 0.0073), favouring the Lab4P mice (Chow-P). Within groups, the spine density decreased in the Chow-C group from 11.37/10 μm ± 0.36 at baseline (BLA) to 9.35/10 μm ± 0.26 (*p* = 0.0001) after 12 weeks and 8.78/10 μm ± 0.28, (*p* < 0.0001) at 24 weeks. For the Chow-P group, spine density was 11.88/10 μm ± 0.40 at 12 weeks and 10.80/10 μm ± 0.50 at 24 weeks. Differential counts of the spines indicated thin spines as the most abundant followed by mushroom and then by stubby (Figure 2iii). At both 12 and 24 weeks of intervention, Chow-C mice displayed fewer mushroom spines with a significant loss of thin spines compared to BLA (*p* = 0.0006 and *p* = 0.0001, respectively). In Chow-P mice after 12 weeks intervention, the density of mushroom (37.3%, *p* = 0.0001) and thin spines (33.2%, *p* = 0.0016) were higher than in Chow-C and were similar densities to the BLA group. The temporal changes in neuronal spine density in chow-fed 3xTg-AD mice over the 12-week study period are shown in Appendix A and are consistent with AD progression.

mRNA expression analysis of the hippocampi of the Chow-P mice (Table 1) indicated significantly lower levels of IL-10 transcripts (66.4%, *p* = 0.0005) after 24 weeks compared to Chow-C, whereas all other genes were unchanged between groups.

### 2.2. Lab4P Maintains Cognitive Performance and Preserves Hippocampal Spine Densities in HFD-Fed 3xTg-AD Mice (STUDY B)

Male mice (*n* = 15) were fed from birth with a standard chow diet for 12 weeks. The baseline group (BLB, *n* = 6) were sacrificed, while the probiotic group (*n* = 4) received a high fat diet (HFD) supplemented with Lab4P (HFD-P) for 12 weeks, and the control group (*n* = 5) received HFD alone (HFD-C) for 12 weeks (Figure 3A). Assessment of NOR in HFD mice (Figure 3Bi) revealed significant differences in DR; between groups, the HFD-P group was higher (65.2%, *p* = 0.0047, Figure 3Bii) than the HFD-C group at 12 weeks. The HFD-P group was (0.76 ± 0.06) comparable to BLB (0.67 ± 0.01), but the HFD-C group decreased significantly over this time (0.46 ± 0.06, *p* = 0.0211). In the OFT, no significant between-group differences were observed for any behaviour at 12 weeks (Figure 3Biii). Distance travelled, rearing and digging were lower in HFD-P compared to HFD-C, trending towards significance for digging at 12 weeks (−54.1%, *p* = 0.0728). In the HFD-P group, at the end of the intervention period, all behaviours remained similar to or dropped from BLB trending towards significance for rearing (−63.2%, *p* = 0.0617). Behavioural patterns in HFD-C were more varied compared with BLB and distance travelled significantly increased from BLB (52.2%, *p* = 0.0109).

At the end of the intervention period, the total number of dendritic spines were reduced in the HFD-C mice (27.5%, *p* = 0.0002, Figure 4i) compared to BLB. In the HFD-P mice, CA1 spine loss was prevented (compared with HFD-C, 27.1%, *p* < 0.0001) and similar to that of BLB mice (11.21/10 μm ± 0.49 vs 12.16/10 μm ± 0.46, respectively, Figure 4ii). Subtyping the HFD-C mice dendritic spines based on morphology revealed significant reductions in all three spine subtypes compared to BLB (stubby *p* = 0.0248, mushroom, *p* = 0.0033, and thin *p* < 0.0001, Figure 4iii). In HFD-P mice, dendritic spines were protected with the density of thin spines comparable to BLB and significantly higher (38.8%, *p* = 0.0037) than HFD-C.

mRNA expression analysis of the hippocampus (Table 2) revealed lower transcript levels of the inflammatory cytokines interleukin (IL)-1β (72.1%, *p* = 0.0015) and TNF-α (43.4%, *p* = 0.0179) in the HFD-P group compared to HFD-C at 12 weeks, whilst all other genes were unchanged between groups. In the HFD-C mice, expression levels of IL-1β and TNF-α increased significantly from BLB (~5.5-fold, *p* < 0.0001), but HFD-P remained closer to BLB. A similar pattern of change was observed in the case of IL-6 where levels were significantly increased from BLB in HFD-C animals (2.1-fold, *p* = 0.0346) while those seen in the HFD-P group remained similar to BLB, albeit with no significant difference between intervention groups.

### 2.3. Lab4P CM Promotes Survival in Human SH-SY5Y Cells Challenged with β-Amyloid

Successful differentiation of SH-SY5Y into a cholinergic phenotype was confirmed by the development of neuritic projections [32,33,34] (Figure 5Ai,Aii) and increased expression of neuronal and cholinergic mRNA markers (Figure 5Aiii): NeuN [35] (1.3-fold, *p* = 0.047), TH [35] (158.7-fold, *p* < 0.001), SLC18A [32] (6.7-fold, *p* < 0.001), CDK5 [32] (1.16-fold, *p* = 0.0209) and PSEN1 [32] (3.5-fold, *p* < 0.001). Exposure to human β-Amyloid 1-42 peptides for 48 h reduced the viability of undifferentiated and differentiated SH-SY5Y cells with maximal toxicity observed in response to 0.1 μM β-Amyloid (Figure 5B), which was used for subsequent neurotoxicity experiments. 

In undifferentiated SH-SY5Y cells pre-incubated with 50% Lab4P CM prior to the inclusion of the β-Amyloid, the survival rate was 58.7% compared to the vehicle control (Figure 5Ci); it was significantly higher (*p* = 0.042) than a survival rate of 48.8% in response to incubation with β-Amyloid alone. In differentiated cells pre-incubated with 5 to 50% Lab4P CM for 24 h prior to β-Amyloid inclusion (Figure 5Cii), survival rates ranged between 70.9 and 74.3% compared to the vehicle control; significantly higher (*p* < 0.001 for all) than a survival rate of 55.6% in response to β-Amyloid alone. The neurite length of differentiated SH-SY5Y cells was assessed and found to be unaffected by incubation with β-Amyloid 1-42 with or without pre-incubation with 50% Lab4P CM (Appendix A).

mRNA expression levels of inflammatory and apoptotic genes were assessed in differentiated SH-SY5Y cells exposed to β-Amyloid with or without pre-stimulation with 50% Lab4P CM (Table 3) and revealed a significant reduction in IL-6 expression in the cell exposed to β-Amyloid plus Lab4P CM compared with the vehicle control (72%, *p* = 0.02). No changes in expression were observed in the expression of IL-8 and TNF-α mRNA; IL-1B and IL-10 mRNAs were undetectable in these cells. There were no statistically significant changes in the Bax:Bcl-2 ratios.

## 3. Discussion

The data presented in this manuscript demonstrate the preventative effects of the Lab4P probiotic on cognitive decline, neurodegeneration and neuroinflammation in the 3xTg-AD mouse model in the presence or absence of a metabolic challenge, whilst in human SH-SY5Y neurones the probiotic reduced β-Amyloid neurotoxicity. Overall these data suggest that the Lab4P probiotic has the potential to play a role in the mitigation of the cognitive decline and neuroinflammation associated with AD-related pathologies.

Cognitive capabilities of rodents, particularly memory and learning, can be assessed using a number of different tests. Here, we used a version of the Novel Object Recognition (NOR) test to generate the discrimination ratios (DRs) of the mice, and declines in performance in the control groups of both the chow- and HFD-fed animals were ameliorated by supplementation with probiotics. The DRs for mice challenged with HFD for 3 months (HFD-C) were found to be lower than those observed for mice fed on standard chow (see Appendix A) indicating that the HFD was worsening disease progression (as observed by other groups [13,14,15]). These outcomes align well with our previous probiotic studies showing reduced cognitive decline in 3xTg mice on a high fat diet [22] and better memory in aged Wistar rats [36]. Other groups have shown probiotic-mediated cognitive improvements with the 3xTg-AD mouse model receiving a multi-strain probiotic [20,37,38] and with other transgenic AD mouse models receiving either a single strain probiotic (*Lactobacillus johnsonii*) [39] or a multistrain (VSL#3) product [40].

Disease progression in 3xTg-AD mice impacts upon general/stereotypical mouse behaviours such as activity levels, exploration and rearing as illustrated by the open field test (OFT) [41,42]. The Chow-C mice showed indications of less distance travelled (exploration) and rearing than the mice in the Chow-P group. One of the characteristic traits that is absent in 3xTg-AD mice is fear of open spaces [41], but it was found that the Chow-C mice spent progressively less time in the centre of the test arena, whereas no changes were observed in the Chow-P group. The metabolically challenged HFD-C mice showed ‘hyperactive’ OFT responses to exploration, digging and rearing not seen in the HFD-P mice. It has been suggested that HFD feeding accelerates the decline of cognition and behaviour in 3xTg-AD mice [43], although reports are conflicting [44]. 3xTg-AD mice receiving the SLAB51 probiotic showed no impact upon ambulatory or stereotypic behaviour in the open field test [20].

Both aging and metabolic syndrome are associated with neurodegeneration typified by deterioration/loss of neuronal synaptic connections, notably, the loss of postsynaptic terminals (dendritic spines) that play an essential role in synaptic signalling and cognitive function [45,46,47] and spine loss is exacerbated in neurodegenerative conditions such as AD [8,48]. Significant losses in neuronal spine densities were observed on CA1 apical dendrites in the hippocampi of both the Chow-C and HFD-C mice, but no changes were observed in the mice receiving Lab4P which protected neuronal spine plasticity and integrity. In comparison with baseline mice (BLA), there was a decline in spine density in the HFD-C mice after 3 months (see Appendix A), again, indicating an acceleration of decline in response to the HFD [13,14,15]. The dendritic spine phenotypes are believed to have distinct functions with mushroom spines, considered important for memory and thin spines involved in learning [49]. The density of all spine categories was significantly decreased in the HFD-C mice whereas only the density of thin spines decreased significantly in the Chow-C mice, again illustrating the impact of HFD. Lab4P preserved the integrity of the mushroom and thin spines in the chow fed mice and for all spine phenotypes in the HFD-fed mice, and we have previously shown a probiotic impact in 3xTg-AD mice fed a high fat diet [22].

Neuroinflammation driving neuronal loss is a hallmark of AD [50,51,52] and we have previously observed indications of anti-inflammatory changes in the brains of probiotic fed 3xTg-AD mice [22]. IL-10 is an anti-inflammatory cytokine associated with the resolution of neuroinflammation [53] and, in the current study, reduced transcript levels of IL-10 were recorded in the hippocampus of the Chow-P mice, possibly indicating a less inflamed state. In contrast, IL-1β, IL-6 and TNF-α (major pro-inflammatory cytokines associated with synaptic loss [54,55] and neuronal death [56]) increased from baseline in the HFD-C mice. Supplementation with Lab4P significantly abrogated these increases, again indicating an anti-inflammatory capability. This was comparable with findings in works involving the SLAB51 probiotic [20] and with 5xFAD-AD mice supplemented with *Lactobacillus salivarius* [57]. It is worth noting that elevated mRNA levels of pro-inflammatory genes, including IL-1β, IL-6 and TNF-α, were not found in the hippocampi of Chow-C mice, suggesting that the generation of the overtly pro-inflammatory state may be linked to the HFD feeding regime. These data support implications of ‘overweight status’ significantly increasing risks of AD later in life [12,58].

Obesity is recognised as an inflammatory disorder driving both systemic and neurological inflammation [59] and Lab4P has been shown to (i) inhibit diet-induced weight gain in wild-type mice on a HFD [27] and (ii) induce significant weight loss in free-living overweight and obese human adults [26,28]. The Lab4P probiotic had no impact upon the extent of the diet-induced weight gain in the mice (Appendix A). Lactic acid (a major probiotic metabolic end product) has been shown to exert anti-inflammatory effects in peripheral blood mononuclear cells [60], and in aging rats receiving probiotics where improvements in learning and memory were found to be associated with increased levels of lactate in the brain [36]. The Lab4P consortium harbours a number of putative genes involved in the generation short chain fatty acids (SCFA) [61] that impact upon plasma SCFA levels in vivo [62]. SCFA are thought to exert anti-inflammatory effects on the brain [63]. 

The accumulation and deposition of β-Amyloid is believed to play a key role in AD pathology in humans and is associated with neurodegeneration and cognitive decline [64]. β-Amyloid deposition was not detected in the hippocampus of our cohort of 3xTg-AD mice [22]—potentially due to the noted phenotypic changes that have occurred in the strain since its generation twenty years ago in 2003 [65,66]. Cholinergic neurones are particularly vulnerable to β-Amyloid [67] which are highly abundant in the hippocampus [68]. We combined human SH-SY5Y neuronal cells (in both their naïve undifferentiated form and after RA/BDNF differentiation into a cholinergic phenotype [32]) with neurotoxic β-Amyloid 1-42 peptides and Lab4P metabolites in order to assess the neuroprotective capabilities of the probiotic. The β-Amyloid exerted significant loss of viability in both naïve and cholinergic SH-SY5Y cells and these effects were abrogated by pre-incubation with Lab4P. In the cholinergic cells, neuroprotection by Lab4P occurred alongside a significant reduction in mRNA levels of the pro-inflammatory cytokine, IL-6, supporting the observations made in the hippocampus of HFD-fed 3xTg-AD mice receiving Lab4P.

Our findings align well with the work of Sirin and colleagues in undifferentiated SH-SY5Y cells demonstrating the protective effects of exopolysaccharides from *Lactobacillus delbrueckii* ssp. *bulgaricus B3* and *Lactobacillus plantarum* GD2 against β-Amyloid 1-42 toxicity [69]. To date, all other probiotic studies using SH-SY5Y cells have worked solely in non-cholinergic phenotypes and have demonstrated neuroprotective effects against a range of toxic challenges [25,70,71,72,73,74]. Our previous and current work indicates that Lab4P can protect undifferentiated SH-SY5Y cells against (i) rotenone (Parkinson’s disease-like neurodegeneration [75]), (ii) serum deprivation (intracellular reactive oxygen species accumulation and apoptosis [76]) and (iii) D-galactose (cellular senescence/aging in vitro [77], see Appendix A).

In summary, this study identified the ability of the Lab4P probiotic consortium to slow the progression of cognitive decline and neurodegeneration in 3xTg-AD mice in the presence or absence of a “metabolic” challenge, and a protective effect against neurodegeneration and neuroinflammation was observed in human SH-SY5Y neurones. Together, our findings support further in-depth assessment of the prophylactic neuroprotective potential of Lab4P in both animal models and in human trials.

## 4. Materials and Methods

### 4.1. Experiments in 3xTg-AD Mice

#### 4.1.1. Probiotic Intervention

The Lab4P probiotic consortium is composed of Lactobacillus acidophilus CUL21 (NCIMB 30156), *Lactobacillus acidophilus* CUL60 (NCIMB 30157), *Lactobacillus plantarum* CUL66 (NCIMB 30280), *Bifidobacterium bifidum* CUL20 (NCIMB 30153) and *Bifidobacterium animalis* subsp. *lactis* CUL34 (NCIMB 30172) and was administered as a lyophilised preparation (mixed with feed) delivering a daily dose of ~5 × 10^8^ colony forming units (CFU)/mouse/day (human equivalent dose of ~5 × 10^10^ CFU/day).

#### 4.1.2. Mouse Husbandry and Study Designs 

Mouse experiments were performed under the UK Home Office Project Licence (PPL P8159A562). The 3xTg-AD mice were bred from an in-house colony, founders originally purchased from JAX laboratories (B6;129-Psen1tm1MpmTg (APPSwe,tauP301L) 1Lfa/Mmjax; Genetic Background: C57BL/6; 129X1/SvJ; 129S1/Sv) (Bar Harbor, ME, USA, strain 004807) after generation by Frank LaFerla (University of California, Irvine, CA, USA). These mice contain three mutations associated with familial Alzheimer’s disease (APP Swedish, MAPT P301L, and PSEN1 M146V). All mice were housed in Scantainer vented cages in a light and temperature-controlled environment (12 h on/off light at 22 °C) and had ad libitum access to diet and water.

A schematic representation of the study design and the schedule of sample collection/analysis is shown in Figure 1A and Figure 3A. Male mice (*n* = 69) were fed from birth with standard chow diet (Teklad diet by Envigo, Appendix A) for 12 weeks before 54 mice were randomly selected to enter Study A and 15 mice were selected to enter Study B. The mice entering Study A received a chow diet for a further 12 weeks when the baseline group (BLA, *n* = 18) was sacrificed; the probiotic group (*n* = 18) received standard chow supplemented with Lab4P (Chow-P) for 24 weeks; and the control group (*n* = 18) received standard chow (Chow-C) for 24 weeks. In Study B, the baseline group (BLB, n = 6) were sacrificed immediately; the probiotic group (*n* = 4) received a high fat diet (HFD) supplemented with Lab4P (HFD-P) for 12 weeks; and the control group (*n* = 5) received HFD alone (HFD-C) for 12 weeks. The HFD comprised 21% (*w*/*w*) pork lard with 0.15% (*w*/*w*) cholesterol (Special Diets Services, Witham, UK; product code: 821,424, Appendix A).

For both studies, food and water intake were monitored and mice were weighed every two weeks. At the end of the study, all mice were euthanized by schedule 1 CO_2_ inhalation and blood was collected via cardiac exsanguination. Mice were cardiac-perfused immediately afterwards with 1 X PBS to ensure clearance of blood from vessels and organs. Brains were removed and micro-dissected to obtain key regions. These subsamples were either analysed immediately (for neuronal spine density) or snap frozen (hippocampi) and stored at −80 °C (for mRNA expression analysis).

#### 4.1.3. Mouse Behavioural Testing

All behavioural testing was performed in a custom-made plastic test arena (39 cm (Height) × 39 cm (Width) × 39 cm (Length)) that was placed in a class II laminar flow hood. Novel object recognition (NOR) testing was performed in three stages over 2 days. On day one, each mouse was placed in the box containing sawdust alone and allowed to explore for 10 min before being returned to its home cage (habituation). The next day, each mouse was again allowed to explore the empty test arena for 10 min (open field test) before being returned to its home cage for 30 min. Mice were returned to the test arena to which two identical objects (familiar objects, FO) had been added for 10 min. Each mouse was again returned to its home cage for 30 min. Finally, one FO was replaced with a novel object (NO) and each mouse was placed in the test arena for a further 10 min. The FOs and NOs were similar in size but differed in colour and shape.

Mouse movements and time spent exploring objects were recorded using a GoPro HERO session camera (GoPro, USA) positioned directly above the test arena, and the data were analysed using Ethovision XT 13 software (Noldus, Wageningen, The Netherlands). The time spent exploring the NO (head oriented towards and within 2 cm) was divided by the time spent exploring both the FO and the NO to provide the discrimination ratio (DR). A reduction in DR denotes less interest in the novel object and implies impaired memory. The analyst was blinded to the group allocation during the test. 

#### 4.1.4. Staining, Imaging and Morphological Classification of Neuronal Dendritic Spines in the Hippocampus

Hippocampal CA1 dendritic spines were determined according to previous descriptions [22,48]. In brief, the fluorometric carbocyanine dye 1,1′-Dioctadecyl-3,3,3′,3′-Tetramethylindocarbocyanine Perchlorate (DiI) (Life Technologies, Carlsbad, CA, USA) was dissolved in dichloromethane and applied dropwise to coat 1.67 µm diameter tungsten particles (Bio-Rad, Hercules, CA, USA). Dye coated tungsten particles were funnelled into ethylene tetrafluoroethylene (ETFE) tubing and cut into ‘bullets’ for ballistic DiOlistic labelling onto fresh hippocampal slices (200 μm thick) at 100 psi using a Helios Gene Gun (Bio-Rad, USA) through a 3.0 µm pore size cell culture insert. Dye labelling was checked under a fluorescence microscope to confirm neuronal labelling with subsequent labelling controlled by adjusting delivery pressure and frequent replacement of inserts. Hippocampal slices were placed in Neurobasal-A medium (Life Technologies) and incubated at 37 °C with 5% CO_2_ for 20 min to facilitate dye diffusion. Slices were fixed in 4% paraformaldehyde (PFA) for 30 min at room temperature, nuclear stained with Hoechst 33342, mounted in FluorSave and stored in the dark at 4 °C until imaging the following day.

Dendritic spines were imaged on secondary dendrites within the striatum radium region of hippocampal CA1 neurons using a Leica SP8 laser-scanning confocal microscope with lightning deconvolution (Leica Microsystems, Milton Keynes, UK). Dendritic segments were imaged under a 63× objective (*z* axis interval 0.2 µm) with images processed under Lightning deconvolution (Leica). Typically, each mouse generated approximately 10–15 slices which corresponded to around 10 labelled CA1 neurons.

Dendrite segments longer than 30 µm were 3D reconstructed in BitPlane Imaris software version 9.3.1 using the module Filament Tracer with default thresholding based around a ‘region of interest’. Dendritic spines were classified based on morphology using the Spine Classifier MATLAB extension. Spines were distinguished on the basis of spine length and spine head size and classified as follows: stubby spines were <0.8 µm in length; mushroom spines were >0.8 µm, but ≤3 µm in length with a spine head diameter greater than neck width; and thin spines were >0.8 µm, but ≤3 µm in length but without a bulbous head. Protrusions >3 µm in length were considered to be filopodia, were infrequently detected and therefore not included in the analysis. Each reconstructed dendrite segment was manually checked to ensure correct spine detections with the operator blinded to the experimental condition during reconstruction and data collection.

#### 4.1.5. mRNA Expression Analysis of Hippocampus

Frozen hippocampal tissues were thawed and homogenised with RiboZol (VWR, UK) for 3 × 20 s in a Fast Prep-24 Bead Beater (MPBIO, Santa Ana, CA, USA) with cooling on ice between each round. RNA was extracted and quantitative PCR (qPCR) was performed as described elsewhere [22] using the gene-specific oligonucleotide primers shown in Appendix A. mRNA expression levels in relation to the untreated controls were determined using 2^−ΔCt^, where ΔCt represents the difference between the threshold cycle (CT) for each target gene and the housekeeping gene (β-actin).

### 4.2. Experiments In Vitro in the Human SH-SY5Y Neuronal Cell Line

#### 4.2.1. Maintenance and Differentiation of SH-SY5Y Cell Cultures

SH-SY5Y cells were maintained in T75 culture flasks (Costar, Cambridge, UK) in DMEM/F12 (Labtech, Heathfield, UK) supplemented with 10% (*v*/*v*) heat inactivated fetal bovine serum (Labtech, UK), penicillin (100 U/mL) and streptomycin (100 U/mL) at 37 °C in 5% CO_2_ and 95% humidity. When cells reached ~80% confluence, they were seeded into 96 well (viability assays) or 24 well (mRNA expression assays) tissue culture plates (Costar, Cambridge, UK) at a density of 5 × 10^5^ cells/cm^2^ and incubated at 37 °C in 5% CO_2_ and 95% humidity for 24 h prior to experimentation, or differentiation followed by experimentation. Differentiation was achieved according to the method of de Medeiros *et al.* [32]; cells were incubated for 1 day in DMEM/F12 with 10% serum followed by 3 days in DMEM/F12 with 1% serum plus 10 μM Retinoic acid (RA) and then 3 days in DMEM/F12 with 1% serum plus 10 μM RA plus 50 ng/mL bone derived neurotrophic factor (BDNF) (Abcam, Cambridge, UK). 

#### 4.2.2. Quantification of Neurite Length

Images of the cells (8-bit grey scale, ×10 magnification) at the approximate centre of the tissue culture wells (1 image per well) were captured using a Leica Flexacam C3 microscope camera attached to a Leica DMi1 inverted microscope (Leica Microsystems, Solms, Germany) and were analysed for neurite length using the manual tracing facility in NeuronJ [32]. Analysis was performed on 5 cells per image that were randomly selected; an overlay reference grid with sequentially numbered (top left to bottom right) 1 mm^2^ squares was overlaid onto the image and the cell closest to the centre of a randomly selected square (using a random number generator (https://www.random.org/, last accessed on 24 February 2023) was analysed. The analyst was blinded to the cell treatments.

#### 4.2.3. mRNA Expression Analysis in SH-SY5Y Cells

Total RNA was extracted and quantitative PCR was performed as described elsewhere [25] using the gene-specific oligonucleotide primers shown in Appendix A. mRNA expression levels in differentiated cells were expressed in relation to undifferentiated cells or the vehicle control and determined using 2^−ΔCt^, where ΔCt represents the difference between the threshold cycle (CT) for each target gene and the housekeeping gene (β-actin).

#### 4.2.4. Generation of Lab4P Conditioned Media (CM) and SH-SY5Y Cell Stimulation 

DeMan–Rogosa–Sharpe (MRS) broth was inoculated with lyophilised Lab4P and incubated anaerobically at 37 °C for 18 h. The culture was centrifuged (2500× *g* for 20 min), washed in phosphate buffered saline (PBS), resuspended at 1 × 10^9^ CFU/mL in a 1:1 mix of Dulbecco’s Modified Eagle’s medium and Ham F-12 medium (DMEM/F12) and incubated without agitation under anaerobic conditions at 37 °C for 5 h. The cells were pelleted (2500× *g* for 20 min) and the supernatant was filtered (0.22 µm, Gibson, Bedfordshire, UK), adjusted to pH 7.4 using 1 M NaOH and supplemented with 100 U/mL penicillin and 100 U/mL streptomycin (Labtech, Heathfield, UK) to provide the sterile conditioned medium (CM). For experimentation, CM was diluted in DMEM/F12 supplemented with 100 U/mL penicillin and 100 U/mL streptomycin and 200 µL or 2 mL were applied per well of 96- and 12-well plates, respectively. Unconditioned DMEM/F12 supplemented with 100 U/mL penicillin and 100 U/mL streptomycin was used as a control for the CM. Human β-Amyloid 1-42 peptides (Abcam, Cambridge, UK) were dissolved in DMSO prior to application to the cells.

#### 4.2.5. Assessment of SH-SY5Y Cell Viability

SH-SY5Y cells were washed with 200 µL of PBS (pH 7.4, 37 °C) before exposure to 100 µL of 3-(4,5-dimethythiazol-2-yl)-2,5-diphenyl tetrazolium bromide (MTT) solution (500 µg/mL in DMEM/F12) for 2 h at 37 °C in 5% CO_2_ and 95% humidity. The cells were washed twice by the repeated addition and removal of 200 µL of PBS (pH 7.4, 37 °C) before the addition of 100 µL of DMSO for 5 min (with occasional agitation by hand). The absorbance at 570 nm was read using a colorimetric spectrophotometer and viability data are expressed as percentage survival compared to the control cells that have been arbitrarily assigned as 100%.

### 4.3. Statistical Analysis

#### Studies In Vivo and In Vitro

All data are presented as the mean ± standard error of the mean (SEM) of the assigned number of mice or independent experiments. The normality of all data was assessed using Q-Q plots. For normally distributed data, values of *p* were determined using Student’s *t*-test, one-way analysis of variance (ANOVA) with Tukey’s post-hoc analysis where homogeneity of variance was met (as determined by the modified Levene’s test) or Brown–Forsythe ANOVA with Dunnett’s T3 post-hoc analysis. Where the data were not normally distributed, values of *p* were determined using the Kruskal–Wallis test with Dunn’s post hoc analysis. All statistical tests were performed using GraphPad Prism (Version 8.2.1). Differences were considered significant when *p* < 0.05.

## Figures and Tables

**Figure 1 ijms-24-04683-f001:**
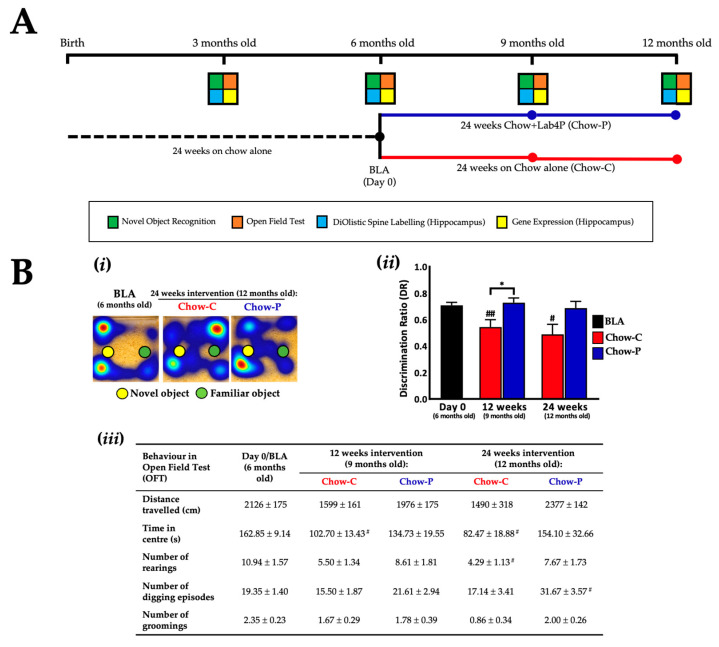
The impact of 24 weeks supplementation with Lab4P on cognition in3xTg-AD mice (Study A). (**A**) 3xTg-AD study design and test/sample collection schedule. Six-month-old male 3xTg-AD mice received a chow diet with or without Lab4P for 24 weeks. Circles indicate analysis time points. BLA, Baseline for study A; BLB, Baseline for study B. (**B-i**) Representative heat maps of mouse movements and (**B-ii**) changes from baseline (BLA) in discrimination ratio (DR) as determined during NOR testing or (**B-iii**) changes from baseline in behaviour during open field testing (OFT) of 3xTg-AD mice receiving chow alone (Chow-C) or chow plus Lab4P (Chow-P) for 24 weeks. Data are expressed as mean ± standard error of the mean (SEM). Mouse numbers for each data point/group were 54 mice (Day 0), 18 mice (12 weeks) and 8 mice (24 weeks). Values of *p* were determined using one-way ANOVA with Tukey’s post hoc analysis, Brown–Forsythe ANOVA with Dunnett’s T3 post hoc analysis or Kruskal–Wallis test with Dunn’s post hoc analysis where * *p* < 0.05 versus the control or # *p* < 0.05 or ## *p* < 0.01 versus the baseline (**B**).

**Figure 2 ijms-24-04683-f002:**
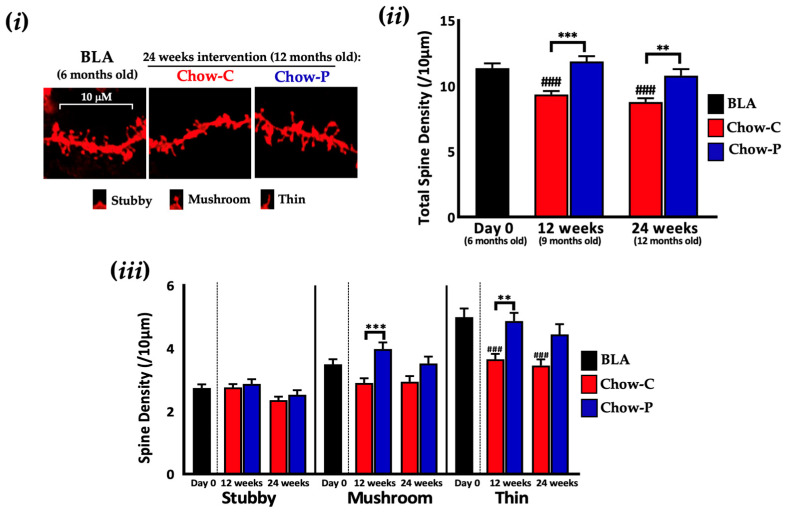
The impact of 24 weeks supplementation with Lab4P on neuronal spine density and mRNA expression in the hippocampus of 3xTg-AD mice (Study A). (**i**) Representative DiOlistic-labelled dendritic segments from hippocampal CA1 apical dendrites with changes from baseline (BLA) in (**ii**) total or (**iii**) stubby, mushroom and thin spine densities of 3xTg-AD mice receiving chow alone (Chow-C) or chow plus Lab4P (Chow-P) for 24 weeks. Data are expressed as mean ± standard error of the mean (SEM). Mouse numbers for each data point/group. Values of *p* were determined using one-way ANOVA with Tukey’s post hoc analysis or Brown–Forsythe ANOVA with Dunnett’s T3 post hoc analysis where ** *p* < 0.01 or *** *p* < 0.001 versus the control or ### *p* < 0.001 versus baseline.

**Figure 3 ijms-24-04683-f003:**
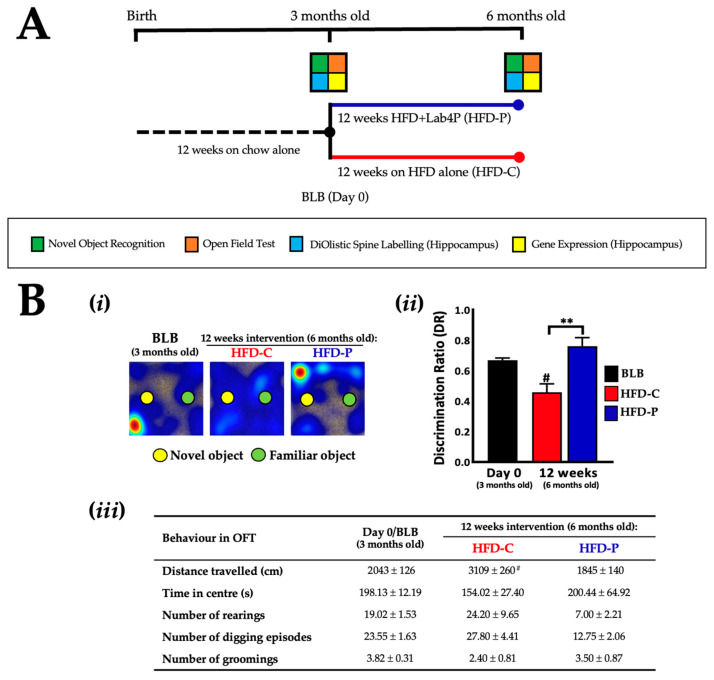
The impact of 12 weeks supplementation with Lab4P on cognition of 3xTg-AD mice on a high fat diet (Study B). (**A**) 3xTg-AD study design and test/sample collection schedule. 3-month-old male 3xTg-AD mice received high fat diet with or without Lab4P for 12 weeks. (**Bi**) Representative heat maps of mouse movements and (**Bii**) changes from baseline (BLB) in discrimination ratio (DR) as determined during Novel Object Recognition (NOR) testing or (**Biii**) changes from baseline in behaviour during open field testing (OFT) of 3xTg-AD mice receiving HFD alone (HFD-C) or HFD plus Lab4P (HFD-P) for 12 weeks. Data are expressed as mean ± standard error of the mean (SEM). Mouse numbers for each data point/group were: 61 mice (Day 0), 5 mice (HFD-C) and 4 mice (HFD-P) (12 weeks). Values of *p* were determined using one-way ANOVA with Tukey’s post hoc analysis or the Kruskal–Wallis test with Dunn’s post hoc analysis where ** indicated *p* < 0.01 versus the control or # *p* < 0.05 versus baseline.

**Figure 4 ijms-24-04683-f004:**
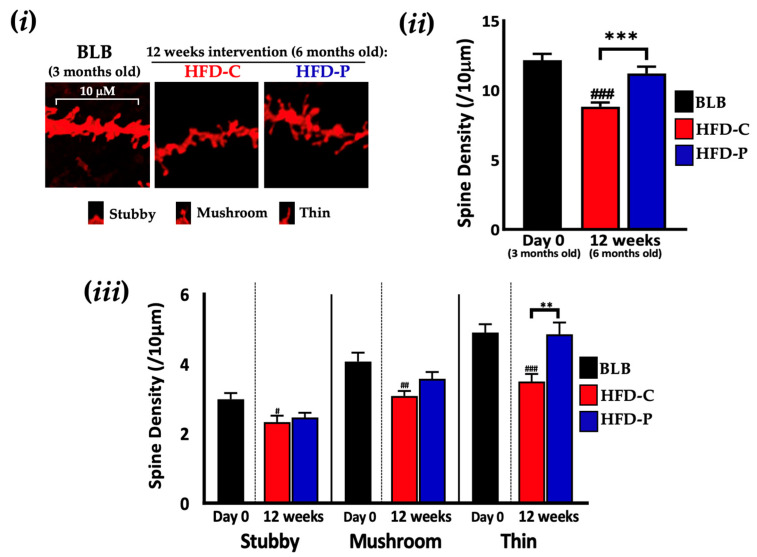
The impact of 12 weeks supplementation with Lab4P on neuronal spine density in the hippocampus of 3xTg-AD mice on a high fat diet (Study B). (**i**) Representative DiOlistic images showing thin, stubby and mushroom dendrites in the hippocampus and changes from baseline (BLB) in (**ii**) total or (**iii**) stubby, mushroom and thin spine densities in the hippocampus of 3xTg-AD mice receiving HFD alone (HFD-C) or HFD plus Lab4P (HFD-P) for 12 weeks. Data are expressed as mean ± standard error of the mean (SEM). Mouse numbers for each data point/group were 4 mice (Day 0) and 4 mice (12 weeks). Values of *p* were determined using one-way ANOVA with Tukey’s post hoc analysis or Brown–Forsythe ANOVA with Dunnett’s T3 post hoc analysis where ** *p* < 0.01 or *** *p* < 0.001 versus the control or # *p* < 0.05, ## *p* < 0.01 or ### *p* < 0.001 versus the baseline. Scale bar 10 µm.

**Figure 5 ijms-24-04683-f005:**
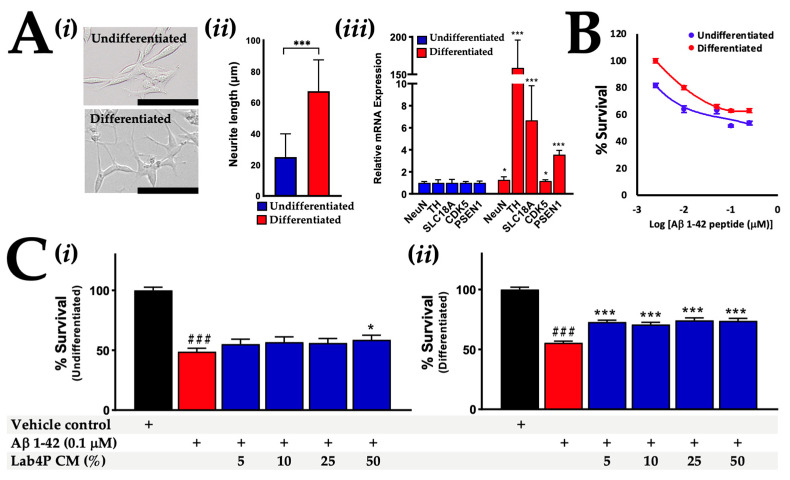
The impact of Lab4P CM on the survival of human SH-SY5Y neurons challenged with β-Amyloid 1-42. (**Ai**) Representative images (scale bar: 100 μm), (**Aii**) neurite length and (**Aiii**) gene expression levels of neuronal and cholinergic markers in undifferentiated and RA/BDNF differentiated SH-SY5Y cells. (**B**) Non-linear regression analysis of the viability of undifferentiated and differentiated SH-SY5Y cells exposed to varying doses of β-Amyloid 1-42 peptides for 48 h. (**C**) The viability of (**Ci**) undifferentiated and (**Cii**) differentiated SH-SY5Y cells that were preincubated with DMEM/F12 or varying doses of Lab4P CM for 24 h prior to the addition of vehicle control (DMSO) or 0.1 μM of β-Amyloid 1-42 peptides for 48 h. Black bar: DMSO control; Red bar: 0.1 μM of β-Amyloid 1-42 alone; Blue bars: 0.1 μM of β-Amyloid 1-42 with Lab4P CM. Gene expression data are expressed relative to undifferentiated cells that have been arbitrarily assigned as 1. Viability is expressed as percentage survival compared to the vehicle control that has been arbitrarily assigned as 100%. The data are presented as the mean ± standard error of the mean (SEM) of at least three independent experiments (with the exception of (**B**) which was a single experiment with quadruple replicates). Values of *p* were determined using students *t*-test (**Aiii**), Kruskal–Wallis test with Dunn’s post-hoc analysis (**Ci**) or one-way ANOVA with Tukey’s post hoc analysis (**Cii**) where * *p* < 0.05 or *** *p* < 0.001 versus undifferentiated cells (**Aii**) or β-Amyloid challenged cells (red bars, **Ci**,**Cii**) and ### *p* < 0.001 versus the vehicle control (black bars, **Ci**,**Cii**). Abbreviations: *NeuN*, neuronal nuclei antigen; *TH*, tyrosine hydroxylase; *SLC18A*, solute carrier family 18 member 2; *CDK5*, Cyclin-dependent kinase; *PSEN1*, Presenilin-1.

**Table 1 ijms-24-04683-t001:** Fold change in mRNA expression in the hippocampus of 3xTg-AD mice (Study A). Shown are changes from baseline in mRNA expression in the hippocampus of 3xTg-AD mice receiving chow alone (Chow-C) or chow plus Lab4P (Chow-P) for 24 weeks (Study A). mRNA expression is presented as a mean fold change ± standard error of the mean (SEM) of five mice compared to baseline that has been arbitrarily set as 1. Values of *p* were determined using one-way ANOVA with Tukey’s post hoc analysis where *** indicated *p* < 0.001 versus the control. Abbreviations: *Iba-1*, ionized calcium-binding adaptor protein-1; *IL*, interleukin; *TNF-α*, tumour necrosis factor-α; *Bcl-2*, B-cell lymphoma 2 gene; *Bax*, *Bcl-2*-associated X apoptosis regulator.

Gene	Baseline (6 Months Old)	12 Weeks Intervention (9 Months Old)		24 Weeks Intervention (12 Months Old)	
		Chow-C	Chow-P	Chow-C	Chow-P
*Iba-1*	1.00 ± 0.23	1.56 ± 0.34	1.22 ± 0.12	1.45 ± 0.18	1.16 ± 0.35
*IL-1β*	1.00 ± 0.25	0.77 ± 0.10	0.84 ± 0.10	1.19 ± 0.22	1.01 ± 0.33
*IL-6*	1.00 ± 0.13	1.08 ± 0.18	0.98 ± 0.19	0.71 ± 0.23	0.67 ± 0.14
*IL-8*	1.00 ± 0.34	0.42 ± 0.05	1.01 ± 0.22	0.56 ± 0.09	0.58 ± 0.13
*IL-10*	1.00 ± 0.10	1.06 ± 0.08	1.18 ± 0.9	1.43 ± 0.26	0.48 ± 0.07 ***
*TNF-α*	1.00 ± 0.18	1.38 ± 0.65	1.11 ± 0.15	2.25 ± 1.05	1.51 ± 0.32
*Bax:Bcl-2*	1.00 ± 0.11	1.06 ± 0.06	1.01 ± 0.04	1.16 ± 0.12	1.30 ± 0.15

**Table 2 ijms-24-04683-t002:** Fold change in mRNA expression in the hippocampus of HFD-fed 3xTg-AD mice (Study B). Shown are the changes from baseline in mRNA expression in the hippocampus of 3xTg-AD mice receiving HFD alone (HFD-C) or HFD plus Lab4P (HFD-P) for 12 weeks (Study B). mRNA expression is presented as a mean fold change ± standard error of the mean (SEM) of five mice compared to baseline that has been arbitrarily set as 1. Values of *p* were determined using one-way ANOVA with Tukey’s post hoc analysis where * *p* < 0.05, ** *p* < 0.01 versus the control or # *p* < 0.05 or ### *p* < 0.001 versus baseline. Abbreviations: *Iba-1*, ionized calcium-binding adaptor protein-1; *IL*, interleukin; *TNF-α*, tumour necrosis factor-α; *Bcl-2*, B-cell lymphoma 2 gene; *Bax*, *Bcl-2*-associated X apoptosis regulator.

Gene	Baseline (3 Months Old)	12 Weeks Intervention (6 Months Old)	
		HFD-C	HFD-P
*Iba-1*	1.00 ± 0.17	0.79 ± 0.19	0.70 ± 0.17
*IL-1β*	1.00 ± 0.15	5.78 ± 1.42 ^###^	1.61 ± 0.36 **
*IL-6*	1.00 ± 0.17	2.09 ± 0.56 ^#^	1.38 ± 0.20
*IL-8*	1.00 ± 0.10	1.45 ± 0.28	1.19 ± 0.13
*IL-10*	1.00 ± 0.08	0.93 ± 0.15	0.67 ± 0.14
*TNF-α*	1.00 ± 0.16	5.58 ± 0.86 ^###^	3.16 ± 0.87 *^#^
*Bax:Bcl-2*	1.00 ± 0.06	1.04 ± 0.08	1.06 ± 0.08

**Table 3 ijms-24-04683-t003:** Fold change in mRNA expression in differentiated SH-SY5Y cells exposed to β-Amyloid 1-42 (0.1 μM) for 48 h, with or without pre-stimulation with Lab4P CM (50%) for 24 h. mRNA expression is presented as a mean fold change ± standard error of the mean (SEM) of four (*IL-6* and *IL-8*) or three (*TNF-α* and *Bax*/*Bcl-2*) independent experiments compared to the vehicle control that has been arbitrarily set as 1. ND: mRNA expression not detected. Values of *p* were determined using one-way ANOVA with Kruskal–Wallis test with Dunn’s post-hoc analysis or Brown–Forsythe ANOVA with Dunnett’s T3 post hoc analysis where # *p* < 0.05 compared to the vehicle control. Abbreviations: *IL*, interleukin; *TNF-α*, tumour necrosis factor-α; *Bcl-2*, B-cell lymphoma 2 gene; *Bax*, *Bcl-2*-associated X apoptosis regulator.

Gene	Vehicle Control	β-Amyloid 1-42 (0.1 μM)	β-Amyloid 1–42 (0.1 μM) + Lab4P CM (50%) Pre-Stimulation
*IL-1β*	ND	ND	ND
*IL-6*	1.00 ± 0.67	1.46 ± 0.91	0.28 ± 0.35 ^#^
*IL-8*	1.00 ± 0.19	1.01 ± 0.29	3.04 ± 1.01
*IL-10*	ND	ND	ND
*TNF-α*	1.00 ± 0.24	0.99 ± 0.71	0.35 ± 0.10
*Bax:Bcl-2*	1.00 ± 0.04	0.97 ± 0.05	1.39 ±0.16

## Data Availability

Not applicable.

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
