# Peer review of "Assessment of Lab4P Probiotic Effects on Cognition in 3xTg-AD Alzheimer’s Disease Model Mice and the SH-SY5Y Neuronal Cell Line"

_ijms, 2023, doi:10.3390/ijms24054683_

Round 1
Reviewer 1 Report
An article entitled "Assessment of Lab4P probiotic effects on cognition in 3xTg-AD Alzheimer's disease model mice and the SHSY-5Y neuronal cell line" gives us more information about how probiotics act on a mouse model of Alzheimer's disease and a neuronal cell line. Testing the effect of the Lab4P probiotic on neurodegeneration, on the cognitive functions of animals, as well as on the levels of expression of pro-inflammatory and apoptotic genes, they showed the possibility of using the studied probiotic as part of the treatment of Alzheimer's disease in the presence or absence of a metabolic challenge. The logic of the research and the objectives of the work are set out clearly and understandably. However, there are a few comments that I would like to address to the authors.
1.The results section should begin with a brief description of studies A and B, giving the experimental design (Figure 1). Now a detailed description of the experiment is in the Methods and Materials section (at the end of the article), and the Experiment Scheme (Figure 1) is at the end of the results section.
2.It is necessary to insert Figures 2-6 in the text of the article, which describes the results presented on them. Figures 2-6 are now at the end of the Results section.
3.Page 15, lines 448-453. “…54 mice were randomly selected to enter Study A and 15 mice were selected to enter Study B. The mice entering Study A received chow diet for a further 12 weeks when the baseline group (BLA, n=10) were sacrificed, the probiotic group (n=18) received standard chow supplemented with Lab4P (Chow-P) for 24 weeks and the control group (n=18) received standard chow (Chow-C) for 24 weeks”. If these numbers are correct, what happened to the 8 mice from Study A?
4.Table 1 does not have a standard error in the third column.
5.Page 5, line 187. Incorrect reference to figure. Replace Figure A6ii with Figure A6iii.
6.Page 3. It is necessary to substantiate the choice of genes for the analysis of their expression in the hippocampus.
7.Apparently, Figure 7 got into the text of the article by accident. There is no explanation for it in any of the sections of the article.
8.There is no general conclusion in the discussion section and no suggestion about the mechanism of action of Lab4P. In this paper, the authors do not investigate it, but provide data from other authors for other probiotics. According to the authors, do all probiotics have the same mechanism of action?
Author Response
- The results section should begin with a brief description of studies A and B, giving the experimental design (Figure 1). Now a detailed description of the experiment is in the Methods and Materials section (at the end of the article), and the Experiment Scheme (Figure 1) is at the end of the results section.
RESPONSE. Thank you for all your feedback. Following your suggestion, we have updated our figures in order to improve understanding of the methodology and data. The schematic of study designs shown in Figure 1 of the original submission has been split into study A and study B that have now been incorporated into the respective data figures (new figure 1 and figure 3). A small summary of each study has also been added in the results section of the revision (lines 88-92 & 18-184, highlighted in yellow). Please note that all figure numbering has been changed throughout the revision in order to accommodate these changes, for example, Figure 2 is now Figure 1. - It is necessary to insert Figures 2-6 in the text of the article, which describes the results presented on them. Figures 2-6 are now at the end of the Results section.
RESPONSE: All figures have now been inserted into the text close to the description of results. - Page 5, lines 448-453. “…54 mice were randomly selected to enter Study A and 15 mice were selected to enter Study B. The mice entering Study A received chow diet for a further 12 weeks when the baseline group (BLA, n=10) were sacrificed, the probiotic group (n=18) received standard chow supplemented with Lab4P (Chow-P) for 24 weeks and the control group (n=18) received standard chow (Chow-C) for 24 weeks”. If these numbers are correct, what happened to the 8 mice from Study A?
RESPONSE: Apologies for this error. There were 18 mice in the BLA group, not 10, and the text has been corrected on line 457of the revision. - Table 1 does not have a standard error in the third column.
RESPONSE: Standard errors were present but not visible due to inadequate column width. We have reduced the font size for all our tables to make sure that all data is visible to read. - Page 5, line 187. Incorrect reference to figure. Replace Figure A6ii with Figure A6iii.
RESPONSE: This error has now been corrected (page 9, Lines 261). Please be aware that Figure 6 has become Figure 5 due to changes made in response to comment #1. - Page 3. It is necessary to substantiate the choice of genes for the analysis of their expression in the hippocampus.
RESPONSE: The relevance of each gene expression marker was described in the discussion section 375 to 389 We focused on inflammatory markers in the hippocampus in order to extend on our previous work in 3xTg AD-prone mice that showed significant anti-inflammatory changes in whole brain extracts of probiotic fed mice [Webberley et al, Frontiers in Neuroscience, 16:843105,2022]. We have added content to the discussion (lines 375-389) to highlight this. - Apparently, Figure 7 got into the text of the article by accident. There is no explanation for it in any of the sections of the article.
RESPONSE: Sincere apologies for this error. The figure was mistakenly inserted into this manuscript and is related to an entirely different project. It has now been removed.
- There is no general conclusion in the discussion section and no suggestion about the mechanism of action of Lab4P. In this paper, the authors do not investigate it, but provide data from other authors for other probiotics. According to the authors, do all probiotics have the same mechanism of action?
RESPONSE: A general conclusion was provided at the end of the discussion section (lines 427 – 432). Whilst we do not yet understand the mechanism of Lab4P action we are aware that the actions of probiotics are species specific and unlikely to occur via the same mechanism of action. Therefore, we felt it important to describe potential mechanisms of action and, in the absence of data relating specifically to Lab4P, used the available literature as a basis. Work into the mechanism of action of Lab4P is currently ongoing.
Reviewer 2 Report
The research article by Thomas S Webberley et al. entitled Assessment of Lab4P probiotic effects on cognition in 3xTg-AD Alzheimer’s disease model mice and the SHSY-5Y neuronal cell line in triple transgenic mice and in SHSY5Y cells. The paper are well presented and written and interesting to the readers however I have the following concerns in the papers.
The authors checked some parameters related to AD, like behaviors, but the authors missed the main markers related to AD, like beta-amyloid, phosphorylated tau, and beta secretased1 in the brain of experimental mice. First of all the authors should examine whether the probiotic treatment to AD mice has any effects on basic markers of AD mice brains.
What is the importance of the inflammatory markers in this study?
The authors checks the AD parameters only;y in the hippocampus but memory is mainly related with the cortex and hippocampus
There are so many probiotics and the authors selected Lab4P for this study what is the reason.
Author Response
- The authors checked some parameters related to AD, like behaviours, but the authors missed the main markers related to AD, like beta-amyloid, phosphorylated tau, and beta secretase-1 in the brain of experimental mice. First of all, the authors should examine whether the probiotic treatment to AD mice has any effects on basic markers of AD mice brains.
RESPONSE: Thank you very much for all your comments. Our previous study conducted on the same cohort of 3xTg mice found no accumulation or deposits of β-amyloid in the brain [Webberley et al, Frontiers in Neuroscience, 16:843105,2022], possibly due to the phenotypic changes which have occurred in this model over the generations since its inception in 2003 by the originating authors (Front Neurosci. 2022 Jan 24;15:785276. doi: 10.3389/fnins.2021.785276 Systematic Phenotyping and Characterization of the 3xTg-AD Mouse Model of Alzheimer's Disease). For this reason, we did not investigate β -amyloid in the current study. We agree that phosphorylated tau and beta-secretase-1 need to be investigated. Experimentally, it is not possible in the current study but will certainly be included in future studies in these mice. - What is the importance of the inflammatory markers in this study?
RESPONSE: Our focus on gene expression markers of inflammation in the hippocampus was based on the findings of our previous work in the same colony of 3xTg AD-prone mice [Webberley et al, Frontiers in Neuroscience, 16:843105,2022] where anti-inflammatory changes were observed in the brain in response to probiotics. The relevance of these markers to Alzheimer’s Disease was covered in the discussion section (lines 375-389). - The authors check the AD parameters only; in the hippocampus but memory is mainly related with the cortex and hippocampus
RESPONSE: We are aware that both the cortex and hippocampus contribute to memory but limited resource confined our analysis to a single region. The hippocampus was selected for analysis on the basis of our previous study in 3xTg mice [Webberley et al, Frontiers in Neuroscience, 16:843105,2022] where probiotic-mediated improvements in novel object recognition occurred alongside the preservation of hippocampal spine densities. - There are so many probiotics and the authors selected Lab4P for this study what is the reason.
RESPONSE: Previous work in vitro demonstrated a neuroprotective role for a related probiotic consortia (Lab4) in challenged SH-SY5Y human neuronal cells [Michael et al, Beneficial Microbes; 10(4): 437-447, 2019]. Furthermore, human studies conducted with Lab4P have indicated that this probiotic may modulate participant-perceived mood scores (Michael et al, Scientific Reports 10;4183, 2020) thus implying an ability to modulate the gut-brain axis. We have expanded the introduction section of the revision (lines 74-75) accordingly.
Reviewer 3 Report
The manuscript entitled “Assessment of Lab4P probiotic effects on cognition in 3xTg-AD Alzheimer’s disease model mice and the SHSY-5Y neuronal cell line” detailed the protective effect of Lab4P. the study was well-planned and implemented.
The manuscript detailed the necessary information and discussed the results properly.
But I have some minor comments on the manuscript.
The composition of Lab4P should be included in the manuscript if no patent or any other restrictions are not there.
The table legend should be at the top of the table.
Correct the typo in the manuscript. E.g., Line 216. Correct the symbol.
Author Response
- The composition of Lab4P should be included in the manuscript if no patent or any other restrictions are not there.
RESPONSE: Thank you for all your feedback. The composition of bacteria in Lab4P was included in the original submission (lines 68-69 of revision). - The table legend should be at the top of the table.
RESPONSE: Thank you for this suggestion. We have amended all tables accordingly. - Correct the typo in the manuscript. E.g., Line 216. Correct the symbol.
RESPONSE: This error has been correct which is now on line 312on the revised manuscript.
Round 2
Reviewer 2 Report
the author solved my all concerned